

# Description and evaluation of NorESM1-F: A fast version of the Norwegian Earth System Model (NorESM)

Chuncheng Guo[1], Mats Bentsen[1], Ingo Bethke[1], Mehmet Ilicak[1], Jerry Tjiputra[1], Thomas Tonniazzo[1], Jörg Schwinger[1], and Odd Helge Otterå[1]

[1]Uni Research Climate, Bjerknes Centre for Climate Research, Bergen, Norway

*Correspondence to:* Chuncheng Guo (chuncheng.guo@norceresearch.no)

**Abstract.**

A new computationally efficient version of the Norwegian Earth System Model (NorESM) is presented. This new version (here termed NorESM1-F) runs about 2.5 times faster (e.g. 90 model years per day on current hardware) than the version that contributed to the fifth phase of the Coupled Model Intercomparison project (CMIP5), i.e., NorESM1-M, and is therefore particularly suitable for multi-millennial paleoclimate and carbon cycle simulations or large ensemble simulations. The speedup is primarily a result of using a prescribed atmosphere aerosol chemistry and a tripolar ocean-sea ice horizontal grid configuration that allows an increase of the ocean-sea ice component time steps. Ocean biogeochemistry can be activated for fully coupled and semi-coupled carbon cycle applications. This paper describes the model and evaluates its performance using observations and NorESM1-M as benchmarks. The evaluation emphasises model stability, important large-scale features in the ocean and sea ice components, internal variability in the coupled system, and climate sensitivity. Simulation results from NorESM1-F in general agree well with observational estimates, and show evident improvements over NorESM1-M, for example, in the strength of the meridional overturning circulation and sea ice simulation, both important metrics in simulating past and future climates. Whereas NorESM1-M showed a slight global cool bias in the upper oceans, NorESM1-F exhibits a global warm bias. In general, however, NorESM1-F has more similarities than dissimilarities compared to NorESM1-M, and some biases and deficiencies known in NorESM1-M remain.

## 1 Introduction

The Norwegian Earth System Model (NorESM, Bentsen et al., 2013) was one of the ~20 models that contributed to the fifth phase of the Coupled Model Intercomparison Project (CMIP5, Taylor et al., 2012). It was built upon the Community Climate System Model, version 4 (CCSM4, Gent et al., 2011) but differs from the latter mainly in the implementation of advanced chemistry-aerosol-cloud-radiation schemes, an isopycnic coordinate ocean model, and the incorporation of the HAMburg Ocean Carbon Cycle (HAMOCC; Maier-Reimer, 1993; Maier-Reimer et al., 2005) model that is adapted to an isopycnic model framework. The basic evaluation and validation as well as transient climate response and future scenario projections of the CMIP5 version of NorESM (NorESM1-M) have been documented by Bentsen et al. (2013) and Iversen et al. (2013).



The capability of fully coupled climate models in performing long integrations without compromising on the model resolution and complexity is always demanding. Paleoclimate simulations, which often require millennial-scale integration to reach equilibrium, usually employ earth system models of intermediate complexity (EMICs) or coupled models with reduced resolution. For example, the low-resolution version of the CCSM4 reported by Shields et al. (2012) employs a horizontal reso-

lution of 3.75° in the atmosphere grid and nominal 3° in the ocean grid, compared with the intermediate (standard) resolution of CCSM4 with a 2° (1°) atmosphere and a nominal 1° ocean grid (Gent et al., 2011). The lower resolution and reduced complexity allow paleoclimate modellers to perform a variety of long simulations including multi-millennial equilibrium and sensitivity experiments.

The NorESM family also features a lower-resolution version (NorESM-L) that was designed for simulations of past cli-

mates (Zhang et al., 2012). NorESM-L employs a similar grid resolution as the lower-resolution CCSM4, and has been used for simulating past climates during, e.g., the mid-Pliocene (Zhang et al., 2012), and the last interglacial periods (Langebroek and Nisancioglu, 2014). However, NorESM-L suffers from a too cold climate, especially in the northern high latitudes, where excessive sea ice associated with a cold surface temperature is found. Such cold bias compromises model credibility in simulating climates when sea ice plays a key role in modulating atmosphere-ocean-sea ice interactions. For instance, during the

last glacial period, millennial scale abrupt climate change prevails as recorded by ice cores from Greenland (Dansgaard et al., 1993). These events, termed Dansgaard-Oeschger (D-O) events, are characterised by a rapid warming from stadial to interstadial states in a matter of a few decades, followed by a gradual cooling to stadial. It has been widely accepted that sea ice cover in the North Atlantic and Nordic Seas exerts a strong control on the Greenland temperature and high latitudes climate (Li et al., 2005, 2010; Dokken et al., 2013). According to these studies, a rapid retreat of sea ice cover in this region, possibly triggered

by ocean subsurface warming (Dokken et al., 2013), can induce rapid Greenland warming that resembles the onset of a D-O event. Therefore, a reasonable simulation of sea ice is of crucial importance in simulating such sea ice-related climate events; neither a too thin sea ice of small extent nor too thick sea ice of large extent is likely to be able to reasonably reproduce the expected sea ice growth/retreat.

The Bjerknes Centre for Climate Research is in the forefront of research on paleo-climates (Zhang et al., 2012; Luo et al.,

2018), carbon cycling (Tjiputra et al., 2016), climate prediction using ensemble forecasting (Counillon et al., 2016) and large ensemble simulations to study fugure cliamte impacts (Mitchell et al., 2017) and constrain projection uncertainties (Bethke et al., 2017). These areas require a model tool that within weeks-to-months can produce millennium-scale simulations, or large ensembles of shorter simulations, with adequate resolution, process representations and climate performance. The newly assembled NorESM1-F, aiming to upgrade the low-resolution NorESM-L to have a similar climate performance as NorESM1-

M, satisfies these qualities, and with its advantage in the speed of integration, NorESM1-F is expected to significantly expand BCCR's capabilities to address the research topics mentioned above. Similar as NorESM1-M, NorESM1-F has a horizontal resolution of ∼2° for the atmosphere and land components and nominal 1° for the ocean and sea ice components. On the new Norwegian infrastructure HPC NeXtScale nx360 architecture "FRAM", a model throughput of ∼90 model years per day is measured with ∼600 cores and ocean biogeochemistry deactivated, whereas with ocean biogeochemistry, the throughput is

reduced to ∼70 model years per day with ∼500 cores.





A number of recent code developments for the next generation of NorESM (i.e., NorESM2 for CMIP6) were implemented in NorESM1-F and will be introduced in the next section. A major improvement in the simulation results is a more realistic representation of the Atlantic Meridional Overturning Circulation (AMOC) in NorESM1-F compared to NorESM1-M. As discussed above on the importance of sea ice simulation in the last glacial, a realistic representation of the AMOC and its

associated heat and freshwater transport is also crucial in simulating climates in the past and future. Marine sediment cores in the North Atlantic have revealed the fluctuations of AMOC in the last glacial cycle (Böhm et al., 2015; Henry et al., 2016), which have been regarded as the leading hypotheses in interpreting D-O cycles (Rahmstorf, 2002; Henry et al., 2016). In climate models, the fluctuations of North Atlantic ocean circulation, either realized by freshwater fluxes or spontaneously occurring, are tightly associated with the change of Greenland temperature that mimics D-O events (Ganopolski and Rahmstorf, 2001;

Menviel et al., 2014; Peltier and Vettoretti, 2014). Such variations of AMOC pose challenges on climate models, and those with a reasonable AMOC representation are best suited for studying paleoclimates, especially for the abrupt climate change events that are tightly associated with AMOC variations. Compared to NorESM1-M and NorESM-L, NorESM1-F shows improved skills in simulating sea ice and AMOC. NorESM1-M has a strong AMOC with high variability (Bentsen et al., 2013), whereas the strength of AMOC is reasonably simulated in NorESM1-F and matches observations-based estimate well.

This paper is devoted to the description and basic evaluation of NorESM1-F, a new model system that is already used in latest paleo and carbon cycle studies (Luo et al., 2018). Section 2 provides a general overview of NorESM1-F version (with focus on model development since NorESM1-M) and experimental design. The model's equilibrium state and stability under constant pre-industrial forcings is assessed in Section 3. Simulated mean states in the ocean and sea ice components are shown and discussed in Section 4. Section 5 focuses on the model internal climate variability. In Section 6, twentieth-century climate

evolution and model climate sensitivity are assessed. The paper is summarised in Section 7.

## 2   Model and experiments

As briefly introduced in the beginning of the paper, NorESM differs from CCSM4 mainly in the following aspects. First, NorESM employs an isopycnic vertical coordinate ocean model, which originates from the Miami Isopycnic Coordinate Ocean Model (MICOM) (Bleck and Smith, 1990; Bleck et al., 1992) but the codes have been largely modified. A complete review

of MICOM modifications in NorESM1-M was presented in Bentsen et al. (2013). Second, modified chemistry-aerosol-cloud-radiation schemes were implemented to the atmospheric component of NorESM which becomes the Oslo version of CAM4 (CAM4-Oslo, Kirkevåg et al., 2013). Thirdly, the HAMburg Ocean Carbon Cycle (HAMOCC) model was implemented and adapted to the isopycnic ocean model of NorESM, and forms the ocean biogeochemistry module.

The NorESM version that contributes to CMIP5, NorESM1-M, was documented by Bentsen et al. (2013) and Iversen et al.

(2013). NorESM1-M has a ∼2° resolution atmosphere and land configuration, and nominal 1° ocean and sea ice configuration. The NorESM1-M version that also includes biogeochemistry, in particular the ocean carbon cycle, is labelled NorESM1-ME (Tjiputra et al., 2013). We will not give a comprehensive introduction for each component of NorESM; the readers are referred



to Bentsen et al. (2013) for a complete overview. Rather, we will document the new implementations and code developments in NorESM1-F compared to NorESM1-M, as well as measures used to increase the model throughput.

## 2.1 NorESM1-F versus NorESM1-M

### 2.1.1 Measures to improve computational performance

In NorESM1-F, the same atmosphere/land grid is used as NorESM1-M, whereas a tripolar grid with a nominal 1° horizontal resolution is used for the ocean/sea ice components in NorESM1-F instead of the bipolar grid in NorESM1-M. Compared to the bipolar grid, the tripolar grid is more isotropic at high northern latitudes and for comparable resolution allows an almost doubled time integration step for the ocean component.

Model complexity in NorESM1-F is reduced by replacing the comprehensive aerosol-cloud process representations of
NorESM1-M with the standard, prescribed aerosol chemistry of CAM4 (as was done in NorESM-L).

The coupling frequency between atmosphere-sea ice and atmosphere-land is reduced from half-hourly to hourly, allowing the use of an hourly base time step for the sea-ice and land components matching the radiative time step of the atmosphere component as well as the baroclinic time step of the ocean component. We further reduced the dynamic sub-cycling of the sea ice from 120 to 80 sub-cycles. Together, these changes provide a model speed-up of 30%, while having a relatively small effect
on the model's climate (see supplementary Fig. S1).

### 2.1.2 Code updates in atmosphere component

In the atmosphere component, a major formulation change for energy updates and energy conservation (EC) is adopted, consistent with that will be used in NorESM2 and in CAM6. EC follows Williamson et al. (2015), and additionally includes the local contribution to enthalpy, $\alpha dp$, by the moist-hydrostatic pressure work under atmospheric moisture changes. The energy
formulation change alone has a very minor impact on the simulations (Williamson et al., 2015), mainly because it only affects intermediate physics states (i.e., partially updated states of the atmosphere after each parameterisation) which are then discarded before the fully (and correctly) updated state is passed to the dynamical core. CAM4 physics does not appear to be sensitive to the small state errors thus introduced. The effect of local hydrostatic pressure work is more sizeable. In magnitude, it is equivalent to the sensible heat exchanged with the surface when water is transferred, and in areas of tropical convection
it can be locally as large as 50 W m$^{-2}$ in the time mean. This helps to maintain mid- and high-level convective available potential energy, and results in deeper convective heating and in a warming of the tropical tropopause, correcting a known bias in CAM4. In terms of mean precipitation, the impact is modest but beneficial with more rainfall over land in the equatorial zone (see supplementary Fig. S2). When coupled with MICOM, EC results in a cooling of tropical SSTs. The seasonal cycle of SSTs in the equatorial Pacific is markedly improved. Interannual variability however is reduced.
The calculation of air-sea fluxes is changed with respect to NorESM1-M and CCSM, in that the COARE-3 algorithm (Fairall et al., 2003) replaces that of Large et al. (1994). The main goal of this change was to improve the evaporation-wind stress relationship, which appears too steep in CAM4 compared to observations. This is achieved (see supplementary Fig. S3).




Moreover, COARE results in beneficial impacts on the simulated precipitation field. Overall, it dries the model, reducing its wet bias and cooling the mid-troposphere and warming the lower troposphere somewhat. Regionally, it mitigates the severe double-ITCZ problem of NorESM, with more precipitation falling on the Equator in the time mean, and matching a reduction north and south of it. The seasonality is also improved, with e.g., a drying over the Indian subcontinent in DJF, and a wetter Monsoon.

COARE's (warming) impact on mean SSTs is very modest, but the redistribution of convective precipitation is accompanied by a change in the wind-stress curl, which leads to increased Ekman pumping in the shallow overturning circulation of the equatorial Pacific. Possibly as a result of this, the period of simulated interannual variability in the Equatorial Pacific is shorter and more peaked than in NorESM1-M.

The calculation of the solar zenith angle for both radiation and albedos follows Zhou et al. (2015), so that a time-step mean

zenith angle is used instead of a centred instantaneous value, allowing for uniform time-average insulation (and reflection). The diagnosed impact of this change on usual "real world" simulations with many other sources of asymmetry, given also the relatively frequent radiation calls in CAM, is very modest.

### 2.1.3 Code updates in ocean component

In the ocean component of NorESM1-M, leapfrog time-stepping is used for the model dynamics, while for computational

efficiency, forward time-stepping was chosen for biogeochemical and age tracers in NorESM1-ME. Due to inconsistent time-stepping of layer thickness and tracers, tracer conservation was unsatisfactory. Thus, in the model presented here, leapfrog time-stepping is used exclusively improving tracer conservation considerably.

Unphysical variability of the ocean barotropic mode was present in high latitude shelf regions in NorESM1-M leading to breakup and ridging of sea ice and subsequently exaggerated sea ice formation. Targeted damping of external inertia-gravity

waves in shallow regions removed this variability and reduced sea ice thickness biases in shelf regions, particularly off the Siberian coast.

One commonly used parameterization to represent oceanic mesoscale eddies in low resolution ocean models is eddy-induced transport that adiabatically tends to reduce available potential energy (Gent and Mcwilliams, 1990; Gent et al., 1995), called GM hereafter. GM introduces an eddy-induced transport proportional to the slope vector of a local neutral surface. As com-

monly done in layered ocean models, the implementation in NorESM1-M uses the slope vector of isopycnic layer interfaces instead of neutral surfaces. This approach is a reasonable approximation to GM in the isopycnic ocean interior, but is profoundly different from GM when a non-isopycnic bulk surface mixed layer is present. In NorESM1-F the eddy-induced transport has been reformulated to use slope vector of neutral surfaces causing stronger upper ocean restratification and associated generally increased SST and reduced mixed layer depths.

The parameterized oceanic eddy diffusivity (Eden and Greatbatch, 2008; Eden et al., 2009) depends on a Richardson number representing local vertical shear and has previously been computed directly from simulated velocity shear. With the implementation of the improved GM mentioned above and thus availability of the slope vector of a local neutral surface, the large scale Richardson number (Visbeck et al., 1997) can be robustly estimated and used in the parameterization of eddy diffusivity. Overall smoother and lower diffusivities are produced and in particular unrealistic large diffusivities in the deep ocean are mitigated

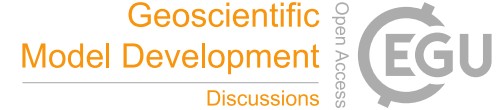



with the use the large scale Richardson number. Further, the eddy diffusivity computation now takes into account the steering level of baroclinic waves. The main impact of this is reduced diffusivity values in the upper ocean that has generally reduced biases in the simulated near-surface temperature and salinity.

The parameterization of mixed layer restratification by submesoscale eddies (Fox-Kemper et al., 2008) has been modified to

make the restratification more efficient at high latitudes.

For diapycnal shear driven mixing, a Richardson number based vertical mixing parameterization based on Large et al. (1994) has been replaced with a more physically sound $k - \epsilon$ model (Umlauf and Burchard, 2005; Ilıcak et al., 2008) that uses a second order turbulence closure. Within the family of $k - \epsilon$ models a new one-equation turbulence closure was developed using the previous Canuto-A stability function (Ilıcak et al., 2008), but parameterized turbulent length scale simply as $l = k^2/\epsilon$ following

Pope (2000). This scheme has been found to provide a satisfactory level of shear driven mixing in the relatively coarse oceanic resolution used in this paper.

The introduction of the above mentioned reformulated GM and $k - \epsilon$ model revealed an issue with the representation of layer thickness at velocity points (staggered with respect to scalar quantities with the Arakawa C-grid used) preventing the representation of realistic vertical velocity shears near steep topography. A new definition of layer thickness at velocity points

has been introduced resolving this issue and ensuring that the $k - \epsilon$ model can provide realistic vertical mixing in gravity currents.

### 2.1.4 Code updates in ocean carbon cycle component

The ocean carbon cycle model coupled to MICOM is based on HAMOCC5 that originated from the work of Maier-Reimer (1993). The HAMOCC model used here has gone through several iterations of development (Maier-Reimer et al., 2005),

including its first adaptation to an isopycnic ocean model (Assmann et al., 2010; Tjiputra et al., 2010). The most recent updates of the model are documented in detail in Tjiputra et al. (2013) and Schwinger et al. (2016). In NorESM1-F, in addition to the updated physical model, the updated version of HAMOCC as described by Schwinger et al. (2016) is employed. The main differences relative to the CMIP5 version (Tjiputra et al., 2013) will be discussed briefly here. As mentioned above, the time-stepping of the biogeochemical tracer fields has been made fully consistent with the leap-frog time-stepping of the physical

fields (see Schwinger et al. (2016) for details). Further, we have activated the advanced particulate sinking scheme based on Kriest (2002), where the sinking speed of particulate organic and inorganic materials are prognostically simulated according to the particle size distributions. The model now includes several preformed tracers, such as oxygen, alkalinity, and phosphate. These preformed tracers, which are set to the respective tracer values at surface, are used to quantify biological and physical carbon pumps in the interior ocean. For air-sea gas exchange computation, the Schmidt numbers have been updated to that of

Gröger and Mikolajewicz (2011).

### 2.2 Experimental design

The strategy and configurations of model experiments follow Bentsen et al. (2013). The fully coupled model was first spun-up for 1000 years to get into a quasi-equilibrium state with a well ventilated upper ocean and little climate drift. The pre-


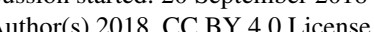


industrial (PI) spin-up used aerosol emissions and concentrations of greenhouse gases defined for year 1850 according to CMIP5 protocols. The solar constant is 1360.9 W m$^{-2}$, and $CO_2$ mixing ratio is set to the pre-industrial value of 284.7 ppm. The ocean component of the model was initialised from rest, and the initial ocean temperature and salinity were from the Polar Science Center Hydrographic Climatology (PHC) 3.0, updated from Steele et al. (2001). For initialization of the ocean

biogeochemical fields, we use the climatological fields from the World Ocean Atlas (WOA, i.e., for oxygen, nitrate, silicate, and phosphate; Garcia et al., 2010a, b) and the Global Ocean Data Analysis Project (GLODAP, i.e., for alkalinity and preindustrial dissolved inorganic carbon; Key et al., 2004).

After the PI spin-up, the simulation was integrated for another 1000 years as the PI control experiment. In this paper, we use the PI control experiment to assess model stability. A historical run was also initialized after the PI spin-up, with observation-

based changes in aerosol, greenhouse gas, volcanic forcing, solar radiation, and land use for the historical time period of 1850-2005 prescribed according to the CMIP5 protocol (for details, see Bentsen et al., 2013). This historical run is used in this paper for comparison with modern observations for both the model mean state and internal variability.

In addition, two idealized CO2 forcing experiments were initialized after the PI spin-up. The first experiment was forced with gradual CO2 increase of 1% per year for 140 years until quadrupling of CO2 relative to the PI level. The second experiment

was forced with abrupt quadrupling of CO2 and was run for 150 years. These two experiments are referred to as "gradual 4×CO2" and "abrupt 4×CO2", respectively, and are used to assess the climate sensitivity of the model.

## 3  Equilibration

In our assessment, the PI experiment reached a satisfactory equilibrium after 1000 years spin-up. The level of equilibration is demonstrated by various representative time series of global mean variables in the control run, e.g., net TOA radiation, T$_{2m}$,

SST, SSS, AMOC strength at 26.5 °N (Fig. 1), and global mean ocean temperature and salinity (Fig. 2).

Efforts were made to achieve a near zero TOA radiation balance during PI spin-up phase. The parameter of minimum relative humidity for low stable cloud formation is tuned to 0.932, compared to 0.90 in NorESM1-M and 0.91 in CCSM4. The increased cloud threshold leads to a lower global mean total cloud amount (52.86%) compared to NorESM1-M (53.76%; both are 1980-2001 average values from the historical run). The mean TOA radiation in the control run has a negative value of -0.04

W m$^{-2}$, with a very small linear trend of 0.02 W m$^{-2}$ over 1000 years that is not statistically significant (statistical significance is tested using the Student's $t$-test with number of degrees of freedom according to Bretherton et al. (1999) that account for autocorrelation; a trend with a $p$ value < 0.05 is considered to be statistically significant). The small negative TOA radiation imbalance leads to a negative net heat flux into the ocean, therefore a cooling of the global ocean is seen (Fig. 2a), with a decrease of 0.07 °C over 1000 years in the control run that is statistically significant. The heat loss mainly occurs in the deep

ocean below 2 km, whereas slight warming is seen below 5.5 km (Fig. 2c). The model experiences very small drifts in the near surface air temperature (-0.11 °C), SST (-0.08 °C) and SSS (-0.04 g/kg over 1000 years; all are statistically significant). The global mean salinity remains stable at a constant value (Fig. 2b), but the upper 4000 m (including SSS) in the ocean experiences a freshening trend, whereas the waters below show a tendency to more saline values (Fig. 2d).




The strength of AMOC shows a small drift during integration in the control run (-0.28 Sv over 1000 years that is statistically significant). The average value and standard deviation of maximum AMOC at 26.5° N are 20.9 Sv and 0.72 Sv, respectively, both are reduced compared to NorESM1-M which features a vigorous AMOC of 30.8 Sv at 26.5° N and a standard deviation of 0.81 Sv.

5 The modelled total sea ice area in the Northern Hemisphere (NH) is close to and slightly larger than present-day climatology with almost no drift in the PI control run (Fig. 3). In the Southern Hemisphere (SH), however, modelled sea ice area is larger than present-day observations both in summer and winter. SH winter sea ice is growing with a linear trend of $1.4 \times 10^6$ km$^2$ over 1000 years that is statistically significant. Furthermore, centennial oscillations (with amplitude up to $3 \times 10^6$ km$^2$) are clearly detected in the SH September (austral winter) sea ice area. Such oscillations are associated with polynyas that occur in 10 the Weddell Sea region. The associated change of sea ice cover regulates the ocean heat transport into the atmosphere and has impact on the SH climate variability (Martin et al., 2015; Pedro et al., 2016). More discussion on the occurrence of Weddell Sea polynyas in NorESM1-F will be given in Section 4.1.

## 4 Modeled large scale features and comparison to observations

With the code update in CAM4, the atmospheric simulation shows certain improvements and reduced bias as described in 15 Section 2.1.2. However, the overall large scale features in NorESM1-M and NorESM1-F are alike. Also given that the ocean and sea ice states are of primary concern to the user community of NorESM1-F, we will therefore not present an evaluation of the atmospheric state. The readers are referred to Bentsen et al. (2013) for the basic evaluation of NorESM1-M and to Section 2.1.2 for the major improvements in NorESM1-F. In this section, we will focus on evaluating the physical and biogeochemical states of the ocean and sea ice components.

20 ### 4.1 Ocean and sea ice state

The large-scale meridional overturning circulation (MOC) in the ocean carries heat and freshwater, and plays an important role in the climate system. Modeled PI global MOC is shown in Fig. 4a. The general structure of the global MOC is similar to NorESM1-M, with a weaker Deacon cell in the Southern Ocean (22 versus 25 Sv) and a stronger counterclockwise deep circulation in the SH (13 versus 10 Sv). However, the clockwise MOC in the NH is evidently weaker than in NorESM1-M 25 (24 versus >30 Sv), which is mainly due to weaker AMOC in NorESM1-F (Fig. 4b). As mentioned in the Introduction, the strength of AMOC (20.9 Sv) was significantly reduced compared to NorESM1-M (30.8 Sv), and reached a level close to the RAPID observations at 26.5°N (∼18 Sv; data from www.rapid.ac.uk/rapidmoc). The upper branch of AMOC in NorESM1-F is also shallower than that in NorESM1-M. Contributing to the reduced AMOC in NorESM1-F is reduced deep convection in the Labrador Sea due to stronger upper ocean restratification by the reformulated GM and modified parameterization of ocean 30 mixed layer restratification by submesoscale eddies.

In NorESM1-M, the strong AMOC carries excessive warm and saline Atlantic waters to the high latitudes, where they are brought to depth and returned southwards in the deep Atlantic. Therefore, NorESM1-M shows a significant warm and saline



bias in the deep Atlantic (see Fig. 14 in Bentsen et al. 2013). With a weakened AMOC in NorESM1-F, the warm and saline bias pattern remains in the deep Atlantic (Fig. 5), but the magnitude of the bias is moderately reduced, indicating an improved representation of water masses in the Atlantic Ocean. Ventilation is also decreased in the deep Atlantic related to the weakened AMOC in NorESM1-F, as revealed by the distribution of ideal age (see supplementary Fig. S4).

Near the ocean surface (in the upper 200 m), the fresh bias also seen in NorESM1-M remains, but the cold bias is replaced by a warm bias now in NorESM1-F. Below the sea surface (200-1000 m), there existed a cold and fresh bias both in the South and North Atlantic in NorESM1-M, whereas in NorESM1-F, the bias pattern and magnitude remains in the South Atlantic but is much mitigated in the North Atlantic. However, a prominent warm and saline bias emerges in the region around 1000 m depth and $30°$N in NorESM1-F which was not seen in NorESM1-M, likely reflecting the incorrect representation of pycnocline

depth in the region. In the Southern Ocean, NorESM1-F shows a fresh and cold bias in the upper 1500 m that is similar to NorESM1-M, and in the lower levels, NorESM1-F shows a reasonable representation of water masses compared to the warm and saline bias seen in NorESM1-M.

A weaker AMOC in NorESM1-F leads to a reduced northward ocean heat transport both in the Atlantic and in the global oceans relative to NorESM1-M. Comparing to estimates constrained by observations, total heat transport is slightly underesti-

mated in both hemispheres (the bias is larger in the SH; Fig. 6a). The ocean heat transport matches observationally constrained estimates well except in the region between 0-20°S where the southward heat transport is underestimated. The slight overestimation of ocean heat transport in the NH is compensated by the slight underestimation of atmospheric heat transport. Both components of heat transport show improvement over NorESM1-M. The underestimated ocean heat transport between 0-20°S is attributed to a too weak southward transport in the Pacific and Indian Ocean and a strong northward transport in the Atlantic

Ocean (Fig. 6b). Furthermore, a northward heat transport of $\sim$0.6 PW is present across the equator (it is nearly zero in observationally constrained estimates), and the surplus comes from the ocean and is mainly due to the (still) excessive northward heat transport in the Atlantic Ocean (Fig. 6b).

Simulated historical (1976-2005 mean) SST and SSS biases are shown in Fig. 7. Global mean SST in NorESM1-F is warm biased (1.09 °C) relative to WOA09, compared to a cold bias of -0.15 °C in NorESM1-M. The warm bias appears in most of

the regions, with the strongest bias seen in the Southern Ocean, eastern boundary upwelling regions, and western boundary current regions (Gulf Stream and Kuroshio), whereas cold biases are found in the western Tropical Atlantic, subpolar Atlantic region, and parts of the Nordic Seas. The cold/warm bias in the subpolar North Atlantic is likely related to the misrepresentation of the North Atlantic Current pathway, i.e., it is extended too far eastward instead of swinging northward off Newfoundland. The presented spatial pattern of SST bias resembles the simulation of NorESM1-M (Bentsen et al., 2013) and CCSM4 (Gent

et al., 2011). However, compared to NorESM1-M, it seems that the much improved representation of AMOC in NorESM1-F does not bring much overall improvement in SST bias in the North Atlantic and Nordic Seas region (the cold bias is reduced while the warm bias is increased in the North Atlantic). The "isolated" strong warm biases in the Gulf Stream and Kuroshio regions indicates that the western boundary currents extend too far north before separating from the coast as is common in ocean models of similar coarse horizontal resolution (Chassignet and Marshall, 2008).



Simulated global mean SSS has a negative bias of -0.51 g/kg, which is larger compared to the bias of -0.15 g/kg in NorESM1-M. Negative SSS bias occurs in most of the world's oceans except in the North Atlantic and off the Siberian coast where strong positive bias is seen (Fig. 7b). The central Arctic is featured with negative bias, as opposed to the positive bias found in NorESM1-M. Furthermore, SSS near the Weddell Sea region features a positive bias, which is likely to be associated with frequent occurrence of polynyas therein whereby waters with higher salinity at depth are able to come to the top.

The distribution of simulated historical (1979-2005) March and September sea ice thickness and extent for both hemispheres is shown in Fig. 8. In the NH, simulated sea ice extent generally follows the observations well but is somewhat underestimated in the Pacific side of the Arctic in both seasons. In the Atlantic side, sea ice extent is slightly less than observations in the Barents Sea and Labrador Sea in March, whereas in September sea ice extent is underestimated along the periphery of Siberian and Alaska coasts. In the SH, sea ice extent agrees well with observations in both seasons and is only slightly underestimated.

Previous NorESM1-M simulated a likely too thick sea ice in both hemispheres (Bentsen et al., 2013). The thickness is reduced in the new NorESM1-F simulation. Modelled winter sea ice thickness in the central Arctic is 1.5-2 m, which is thinner compared to the observed climatology from submarines over years 1975-2000 (Rothrock et al., 2008), and is comparable with satellite observations over years 2003-2008 (Kwok et al., 2009). Given the more rapidly declining trend of sea ice thickness since the 90's (Kwok and Rothrock, 2009), modelled sea ice thickness is deemed to be somewhat thinner compared to observations. In addition, the reported thick sea ice bias off the East Siberian coast in NorESM1-M, that was caused by unphysical oceanic variability in high latitude shelf regions, is significantly improved in NorESM1-F.

In the SH, a noteworthy feature of modeled September sea ice distribution is the frequent emergence of polynyas in the Weddell Sea region. Satellite observations discovered a large Weddell Sea polynya of the size $2-3\times10^5$ km$^2$ in 1974 that persisted in the following two winters (Carsey, 1980). CMIP5 models revealed that Southern Ocean polynyas are common under pre-industrial conditions but cease under anthropogenic climate forcing due to surface freshening (de Lavergne et al., 2014). In NorESM1-F, while the occurrence of Weddell Sea polynyas seems to be stochastic and irregular in both PI control and historical runs, two events of polynyas are cleanly captured during the PI spin-up (see supplementary Fig. S5), each lasting for several decades. Before the opening of the polynya, ocean heat in the model is observed to gradually accumulate from below 4 km and propagate upwards in the deep ocean; then at a tipping point, the whole water column is destabilized and heat is brought up to the ocean surface, where it melts the sea ice and deep convection initiates. The decrease of sea ice area is dramatic, i.e., $\sim3\times10^6$ km$^2$ in the Weddell Sea polynya region (30°W-30°E, 55°S-70°S), which is about one magnitude larger than the observed one in the 1970s. The production of AABW is enhanced during the deep convection process, with an increase in volume transport of $\sim2.5$ Sv at 30°S in the South Atlantic.

## 4.2 Ocean carbon cycle

In the last 100 years of the PI control run, most of the biogeochemical fields in the water column are in quasi equilibrium states (note that for the sediment tracers to reach equilibrium, a much longer spin up integration is required). The global mean net primary production (NPP) and export production are 26.8±0.3 and 4.7±0.1 PgC/yr, respectively. These values are lower than observational estimates from remote sensing as well as relative to the previous model versions (e.g., Behrenfeld and





Falkowski, 1997; Tjiputra et al., 2013). Compared to the remote sensing estimates, the NPP in the model is lower because it fails to resolve the high productivity coastal regions and too low productivity in the oligotrophic subtropical oceans (Fig. 9). The latter is attributed to the too low nutrient supply from subsurface, potentially related to the warm bias and too strong stratification (Schwinger et al., 2016). Fig. 9 also shows that the model is able to simulate the high productions in upwelling

regions of Equatorial Pacific and eastern boundary upwelling systems. Despite relatively low export production, the air-sea gas exchange of CO2 is very close to balance with small outgassing of 0.03±0.06 PgC/yr. Over the historical period the model simulates the expected evolution of oceanic carbon sinks that correspond to higher atmospheric CO2 concentration. Similar to NorESM1-ME (Tjiputra et al., 2013), large increase in uptake rates is pronounced after the year 1950. The simulated increasing $CO_2$ uptakes in the 1980s and 1990s are consistent with the observational-based estimates (see supplementary Fig. S6).

Fig. 10 depicts the statistical performance of the phosphate, oxygen, dissolved inorganic carbon, and alkalinity climatology fields (averaged over the last 30 years of the PI control integration), relative to the respective observational estimates, in the form of Taylor diagrams (Taylor, 2001). Also shown are values from NorESM1-ME (Tjiputra et al., 2013). For these four parameters, it is clear that the current model performance has improved noticeably from the last version, as indicated by the lower parameter biases and higher spatial correlations. These improvements are especially pronounced in the interior ocean as

seen in the phosphate and oxygen tracers (except at 3000 m depth). In the previous model version, interior biases are related to the too strong overturning circulation (Bentsen et al., 2013), and too strong oxygen consumption for biological remineralization. These biases are now reduced.

## 5   Climate variability

In this section, we evaluate two aspects of inter-annual internal variability that are most important to the coupled climate

system: the tropical El Niño-Southern Oscillation (ENSO) mode and the extra-tropical annular modes in both hemispheres.

### 5.1   The ENSO mode

To evaluate ENSO variability, we analyse the long integrations (e.g., the last 500 years) of the PI control experiments of NorESM1-F and NorESM1-M.

Monthly SST and standard deviation of monthly SST anomalies averaged over the NINO3.4 region (bounded by 120°W-

170°W and 5°S-5°N) are shown in Fig. 11a,b) for NorESM1-F, with comparison to HadISST observations and NorESM1-M. NorESM1-M simulates a lower NINO3.4 SST and a higher variability compared to observations all year round, whereas NorESM1-F simulations feature higher NINO3.4 SST (see also the SST bias shown in Fig. 7) and weaker variability (except in September) compared to observations. Maximum NINO3.4 variability in NorESM1-F is achieved in October rather than December as in observations and NorESM1-M. The model skewness for NorESM1-F and NorESM1-M in general shows

opposite sign with observations (except for October-December in NorESM1-M), with the former showing larger nonlinearity in ENSO (Fig. 11c).





The frequency spectrum of the normalized time series of simulated detrended monthly NINO3.4 SST anomalies (Fig. 12a) shows that NorESM1-F has a narrower peak at higher frequency compared with NorESM1-M and observations.

Fig. 12b,c) shows the composite anomalies of DJF surface temperature during El Niño years for NorESM1-F, with a comparison with NorESM1-M. An El Niño year is defined here as a year with the NINO3.4 SST anomalies greater than $1.5\sigma$ ($\sigma$ is standard deviation of NINO3.4 SST anomalies) for three consecutive months, with at least one DJF months. Compared to the temperature anomalies in NorESM1-M, NorESM1-F exhibits a much weaker band of SST anomalies in the NINO 3 and 4 regions; the band also extends further westward in NorESM1-F. The U-shaped negative SST anomalies surrounding the NINO regions are similar between the two experiments, with the anomalies slightly larger in the North Pacific in NorESM1-M. In the Indian Ocean, both models show cooling in the central and eastern part and warming in the rest majority part of the ocean during El Niño years.

## 5.2 Northern and Southern Annular Mode

The Northern Annular Mode (NAM; also known as the Arctic Oscillation) is the leading variability mode in the NH on time scales from days to decades (Thompson and Wallace, 2000). The NAM is defined here as the first empirical orthogonal function (EOF) of the NH (20-90° N) winter (December-February) sea level pressure (SLP) anomalies. The NAM pattern in the NorESM1-F historical experiment is shown in Fig. 13 together with the pattern derived from the NCEP-2 data (Kanamitsu et al., 2002). NorESM1-F generally captures the spatial pattern of NAM, but also inherits similar deficiencies from NorESM1-M, e.g., NAM is stronger over the Arctic and the simulated center is migrated too far east from around Iceland to the Kara Sea, whereas simulated center of action in the North Atlantic are shifted to the east with a less symmetrical structure; the center of action over Pacific is too strong relative to the NCEP-2 data. The SLP variance explained by NAM in NorESM1-F (29%) is stronger than that in NCEP-2 data (22%), but is weaker than that in NorESM1-M (36%).

The Southern Annular mode (SAM, also known as the Antarctic Oscillation) dominates the middle to high latitudes of the SH climate variability. It is defined here as the first EOF of the SH (90-20° S) monthly SLP anomalies. Simulated spatial pattern of SAM (Fig. 13) agrees with that derived from NCEP-2 data in terms of the amplitude of the low pressure anomalies over Antarctica, but the gradient is larger in the Pacific side; there are also some small discrepancies in amplitude and zonal asymmetry of the high pressure anomalies. The SLP variance explained by SAM in NorESM1-F (22%) is close to that in NCEP-2 data (25%).

## 6 Climate evolution of twentieth-century and climate sensitivity

Fig. 14 shows the time series of global mean surface temperature ($T_{2m}$) from the historical run and HadCRUT4 observations (Morice et al., 2012). In general, the model results follow observations and capture well the recent trend of global warming. However, several discrepancies exist between the model run and observations. First, the model seems to overestimate the cooling effects of volcanic activities, with the historical run showing larger drops in temperature following eruptions of Krakatoa (1883), Agung (1963), and the more recent eruption of Pinatubo (1991) than seen in the observational global mean surface





temperature reconstructions. A similar mismatch was also reported for CCSM4 (Gent et al., 2011), suggesting that CAM4 may be overly sensitive to stratospheric volcanic forcing. However, further back in time the global mean surface temperature reconstructions are subject to large uncertainties, and several historical eruption events coincided with El Niño events partly masking the volcanic cooling signal (e.g., McGregor and Timmermann, 2011). Second, the early warming period observed
between 1920s and 1940s is not captured by the model, which is an issue that also existed in NorESM1-M. A recent study, however, demonstrated that a large fraction of this early warming - which was strongest in the Arctic - can be recovered if the Pacific climate variability of the model is synchronised with the observed variability (Svendsen et al., 2018). Third, while the observed more rapid global warming since 1970s is reasonably reproduced by the model, the global warming "slowdown" since 1998 is not captured; rather, the model exhibits a rapid and consistent warming between 1998 and 2005. We note that dis-
crepancies in the interannual temperature can be attributed to differences in the simulated short-term internal climate variability relative to the real world (e.g., ENSO variability), which is expected in coupled climate models.

    Fig. 15 shows the time series of NH and SH sea ice area in March and September for the historical run with comparison to satellite-based estimate of NSIDC (Fetterer et al., 2016). Simulated mean sea ice area agrees with satellite estimates between 1979 and 2005, with slight underestimation over summer in respective hemispheres. NorESM1-F exhibits a declining trend of
NH summer sea ice area after about 1960 and a rapid decline in the last decade (1996-2005). The latter rapid decline is also seen in the satellite estimates. The historical simulation shows improvement over that of NorESM1-M; the latter simulated a small declining trend in summer sea ice area and thus a delayed ice melting in the last decade. The aforementioned simulated too thick sea ice in NorESM1-M is more resistant to summer melting and contributed to the delayed melting. In the SH, larger sea ice area is simulated in March (austral summer) compared to observations, whereas in September (austral winter), modeled
sea ice area exhibits large variability, with an increasing trend seen in the 1980s and 1990s that agrees with NSIDC estimates.

    Based on the two idealized $CO_2$ forcing experiments, we evaluate the climate sensitivity of NorESM1-F following Iversen et al. (2013) for the evaluation of NorESM1-M. Equilibrium climate sensitivity (ECS), defined as the global change of equilibrium surface air temperature in response to a doubling of atmospheric $CO_2$ concentration, is not available as it requires several thousands years to get a fully ventilated ocean. Instead, we apply the linear regression method of Gregory et al. (2004) to
approximately estimate ECS (Fig. 16). In the "abrupt $4\times CO_2$" experiment, it is assumed that the change of TOA radiation flux $\Delta R(t)$ (W m$^{-2}$; t is time) is linearly dependent on the change of global mean surface air temperature $\Delta T(t)$ (°C); the regression slope $\lambda$ (W m$^{-2}$ °C$^{-1}$) is the climate feedback parameter, and the intercept $\Delta T$ at $\Delta R=0$ divided by two is the estimated ECS in the model. With this method, estimated ECS in NorESM1-F is 2.29 °C, which is lower than the ECS in NorESM1-M (2.87 °C), and is close to the lower bound of a range of CMIP5 models (2.1-4.7 °C) examined by Andrews et al. (2012).
The above estimation of ECS does not take into account the rapid adjustments of the system in the beginning of the "abrupt $4\times CO_2$" experiment and underestimates the instantaneous forcing of the $CO_2$ change (Andrews et al., 2012; Iversen et al., 2013). Rather, following Murphy (1995), an effective climate sensitivity, defined as $ECS_{eff}=\Delta T(t)R_f/(R_f-\Delta T(t))$, is calculated, assuming that the external forcing and the feedback processes are constant during equilibration. The radiative forcing $R_f$ is assumed to be 7.0 W m$^{-2}$ as estimated by Kay et al. (2012). $ECS_{eff}$ is expected to be constant over time if a linear
relationship between $\Delta T(t)$ and $\Delta R(t)$ is assumed, but this is hardly the case due to some slow feedback processes in the model.



$ECS_{eff}$ estimated over the last 40 years of the 150-year long "abrupt $4\times CO_2$" experiment is averaged and yields a mean value of $ECS_{eff}$=2.49 °C. The estimated $ECS_{eff}$ is smaller than that in NorESM1-M (2.86 °C).

Transient climate response (TCR) is also estimated to evaluate model climate sensitivity associated with gradual change of $CO_2$. In the "gradual $4\times CO_2$" experiment, TCR is calculated as the difference of global mean surface temperature between

the time when atmospheric $CO_2$ concentration is doubled (averaging between years 60-80) and the same time in the PI control run. The effective TCR ($TCR_{eff}$) can be similarly derived following the estimate of $ECS_{eff}$. The estimated TCR and $TCR_{eff}$ in NorESM1-F are 1.33 °C and 1.56 °C, respectively; the former is comparable to the estimate in NorESM1-M (1.39 °C), whereas the latter is smaller than that in NorESM1-M (2.32 °C).

## 7  Conclusions

A computationally efficient configuration of NorESM, named NorESM1-F, is introduced and evaluated against observations and the CMIP5 version of NorESM (NorESM1-M). NorESM1-F is designed for millennium-scale and large ensemble simulations, and it aims to upgrade NorESM-L to a version that is comparable with NorESM1-M in terms of model resolution, process representation, and climate performance.

In this paper, we presented a 2000-year long PI simulation, a historical simulation, and two idealised $CO_2$ forcing experi-

ments with gradual CO2 increase of 1% per year until quadrupling and with abrupt quadrupling of $CO_2$ forcing. We assessed the model stability, mean model states (ocean, sea ice, and carbon cycle), model internal climate variability, and model climate sensitivity.

The model reaches satisfying quasi-equilibrium after PI spin-up, with modest long term drift in the subsequent control run. There is a small negative TOA radiation balance and an associated cooling trend of global mean ocean temperature (0.07 °C

over 1000 years), in contrast to NorESM1-M that features a warming (and larger) tendency of 0.13 °C over 500 years. Surface $T_{2m}$, SST, and SSS fields are all reasonably well equilibrated with small tendencies.

A major improvement of NorESM1-F over NorESM1-M is the simulation of AMOC, as the latter features a too strong overturning (20.9 versus 30.8 Sv of the maximum AMOC). The more realistic simulation of AMOC improves the ocean and atmosphere heat transport in the Atlantic Ocean, and reduces the warm and saline bias in the deep Atlantic as simulated in

NorESM1-M. As a consequence of the more realistic overturning circulation, the simulated interior ocean biogeochemical tracers are considerably improved relative to the observations. However, the improved AMOC does not lead to a notable improvement of SST bias in the North Atlantic and Nordic Seas region. Another improvement in NorESM1-F over NorESM1-M is the reduced sea ice thickness off the Siberian coast and the over the Arctic in general.

The simulation of SST and SSS fields is degraded in NorESM1-F compared to NorESM1-M. NorESM1-F shows a global

mean warm bias of 1.09 °C, in contrast to NorESM1-M that has a cold bias of -0.15 °C. The spatial patterns of bias remain similar in the two models. Simulated global mean SSS is downgraded from a bias of -0.15 g/kg in NorESM1-M to -0.51 g/kg in NorESM1-F.

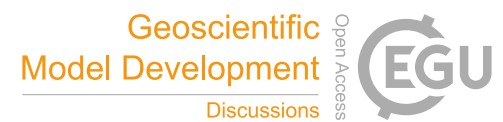

In CAM, several aspects of code updates are implemented in NorESM1-F, resulting in certain improvements as described in Section 2.1.2. The overall large scale features in NorESM1-M are similar to NorESM1-F, and therefore not presented in detail in this work.

In the ocean carbon cycle, simulation of phosphate, oxygen, dissolved inorganic carbon, and alkalinity in NorESM1-F shows
overall noticeable improvements over NorESM1-ME. Considerable improvements are simulated for these tracers in the interior ocean below the mixed layer depth, which are attributed to the better representation of the physical circulation. The observed surface primary production pattern is well reproduced, with the exception of the too low productivity in the coastal regions and subtropical gyres. Productivity in the high latitude during winter period at the respective hemispheres is also too low, a common caveat in CMIP5 models (Nevison et al., 2015).

In the simulation of ENSO, NorESM1-F shows higher NINO3.4 SST and lower variability compared to NorESM1-M and observations across the whole year (except in September). The power spectrum of the NINO3.4 index exhibits a narrower peak at higher frequency in NorESM1-F compared to NorESM1-M and observations. The simulated NAM and SAM in NorESM1-F generally capture the spatial pattern of reanalysis data, but NAM shows (as in NorESM1-M) a stronger center of action in the Arctic located over the Kara Sea and a too strong center of action over the Pacific. The simulated center of action in the North
Atlantic is shifted eastward compared to reanalysis data.

Finally, simulated twentieth-century evolution of global mean surface temperature and sea ice area are in good agreement with observations in NorESM1-F. The experiment likely overestimates the cooling after volcanoes, and does not capture the early warming period between 1920s and 1940s and the recent global warming "slowdown" starting from the end of the last century. Estimation of climate sensitivity using different methods in NorESM1-F shows that the model features a lower climate
sensitivity compared to NorESM1-M and is among the lowest compared to other CMIP5 models.

The model stability and efficiency of NorEMS1-MF are promising, e.g., for multi-millennial paleoclimate simulations. Experiments for selected periods in the past, such as the last interglacial (∼130-115 ka BP) and Marine Isotope Stage 3 (∼60-25 ka BP) at 38 ka BP have already produced promising results (e.g., Luo et al., 2018). Such paleo simulations give us the opportunity to perform data-model comparisons which allow us to further evaluate and quantify the model fidelity, internal
feedbacks, and model climate sensitivity.

*Code and data availability.* The model code can be obtained upon request. Instructions on how to obtain a copy are given at $https : //wiki.met.no/noresm/gitbestpractice$. The full set of model data will be made publicly available through the Norwegian Research Data Archive at $https : //archive.norstore.no$ upon publication.

*Competing interests.* The authors declare that they have no conflict of interest.



*Acknowledgements.* C. Guo and M. Bentsen acknowledge the Ice2Ice project that has received funding from the European Council under the European Community's Seventh Framework Programme (FP7/2007-2013)/ERC grant agreement No. 610055. J. Tjiputra acknowledges Research Council of Norway funded project ORGANIC (239965), and funding from the Bjerknes Centre for Climate Research (BIGCHANGE). J. Schwinger was supported by the Research Council of Norway through project EVA (229771). I. Bethke was supported by the projects BFS
5    BCPU and RCN INES (270061). The simulations were performed on resources provided by UNINETT Sigma2 - the National Infrastructure for High Performance Computing and Data Storage in Norway (nn4659k, ns4659k, nn2345k, ns2345k). Data from the RAPID-WATCH MOC monitoring project are funded by the Natural Environment Research Council and are freely available from www.rapid.ac.uk/rapidmoc





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





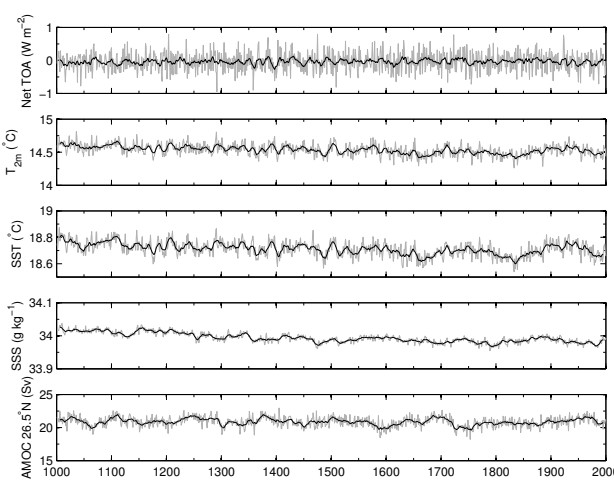

**Figure 1.** Annual mean time series of global mean (from top to bottom) TOA net radiation, $T_{2m}$, SST, SSS, and AMOC strength at 26.5 °N in the PI control run. The grey lines are annual means, whereas the black ones are 10-year running means.





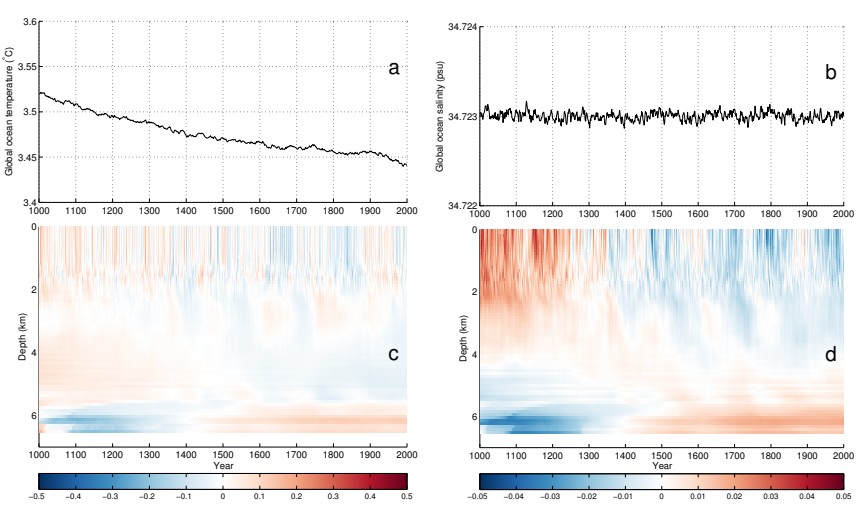

**Figure 2.** Annual mean time series of global ocean mean (a) temperature and (b) salinity. Time evolution of vertical profiles of global mean (c) temperature and (d) salinity anomalies (relative to 1000 year mean of the PI control integration).





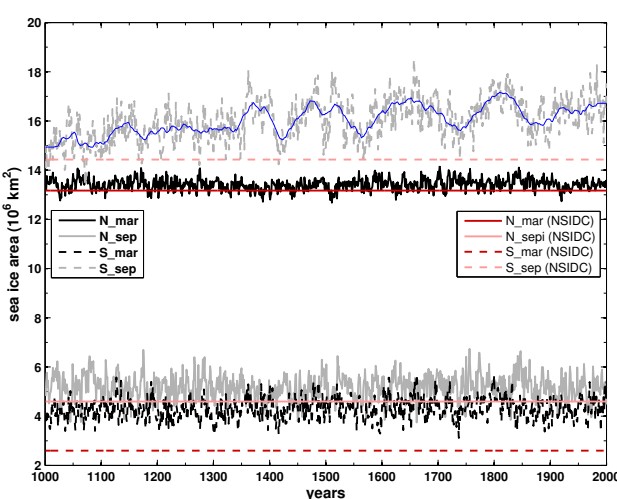

**Figure 3.** Time series of sea ice area ($10^6$ km$^2$) in the Northern and Southern Hemispheres for September and March. Black and grey lines are from the PI control experiment, and red lines are from NSIDC observations (Fetterer et al., 2016). The blue line is the 30-year running mean of the simulated Southern Hemisphere September sea ice area.





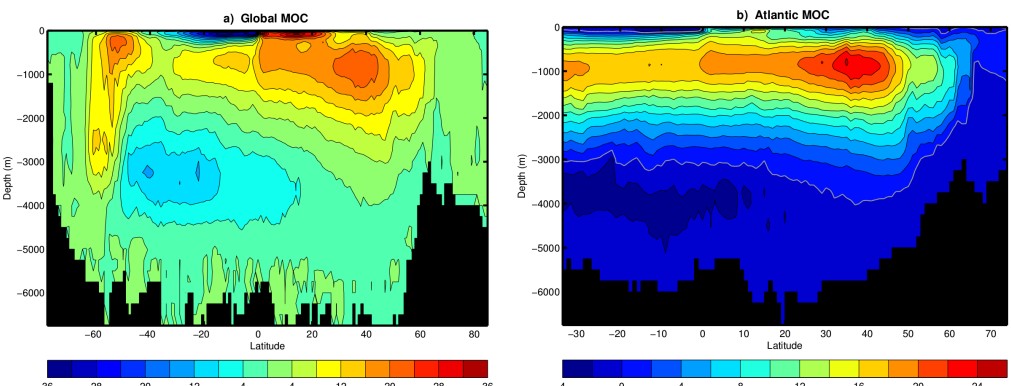

**Figure 4.** Averaged Stream functions (Sv) of a) Global MOC and b) AMOC in depth space in the PI control run.





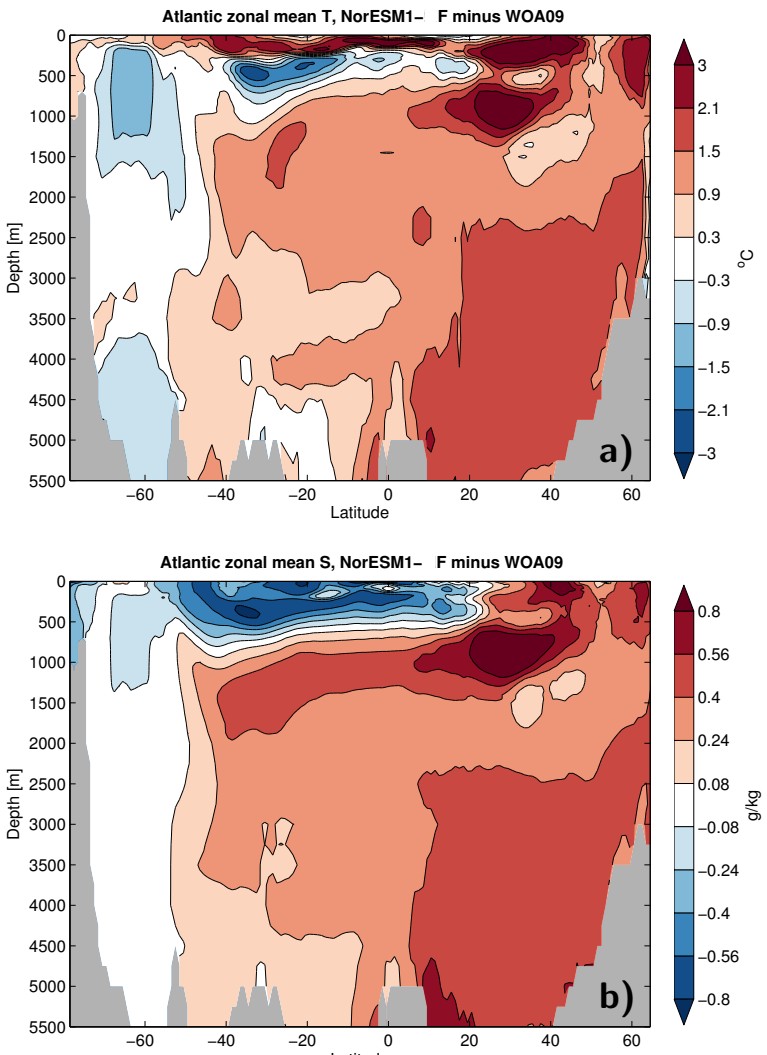

**Figure 5.** Simulated Atlantic zonal mean a) potential temperature and b) salinity biases (historical simulations minus WOA09 data; model results are averaged between years 1976-2005).





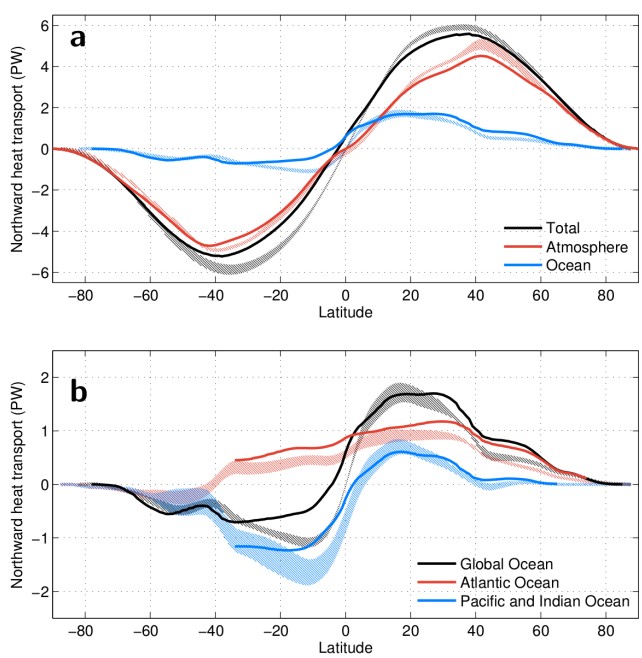

**Figure 6.** Simulated historical (1976-2005 year mean) northward heat transport for a) global atmosphere, ocean, and total, and b) global ocean, its decomposed Atlantic Ocean and Pacific and Indian Ocean. The corresponding hatched areas with uncertainties are estimates from Fasullo and Trenberth (2008). In the model estimation, the ocean heat transport is calculated directly from the ocean model, and the atmospheric heat transport is derived by meridional integration of the difference between the zonal integration of the net TOA and surface heat flux.

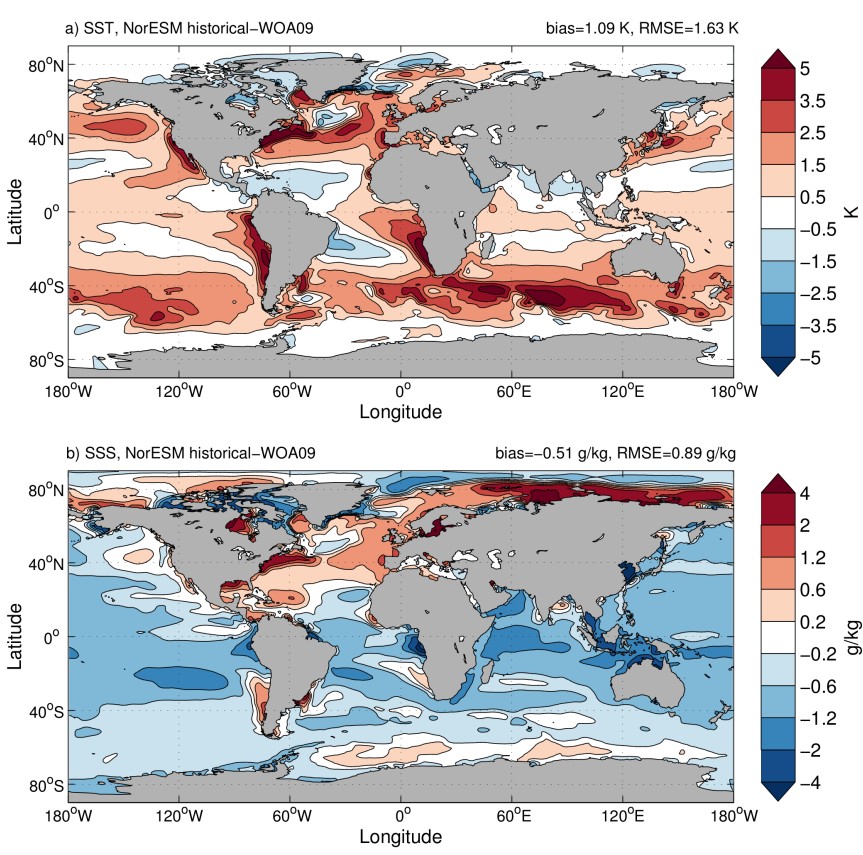

**Figure 7.** Simulated historical a) SST and b) SSS biases (model minus WOA09 data; model results are averaged between years 1976-2005).





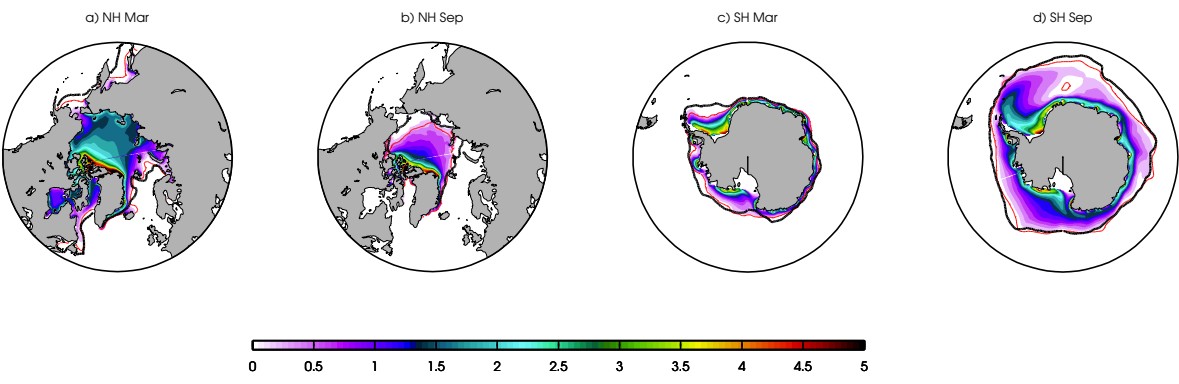

**Figure 8.** Simulated historical sea ice thickness (shading; m) averaged over years 1976-2005 for a) NH March, b) NH September, c) SH March, and d) SH September. The red lines show the modelled mean 15% sea ice concentration, and the thick black lines show the same from the Hadley Centre Sea Ice and Sea Surface Temperature data set (HadISST; Rayner et al., 2003) for the same period.



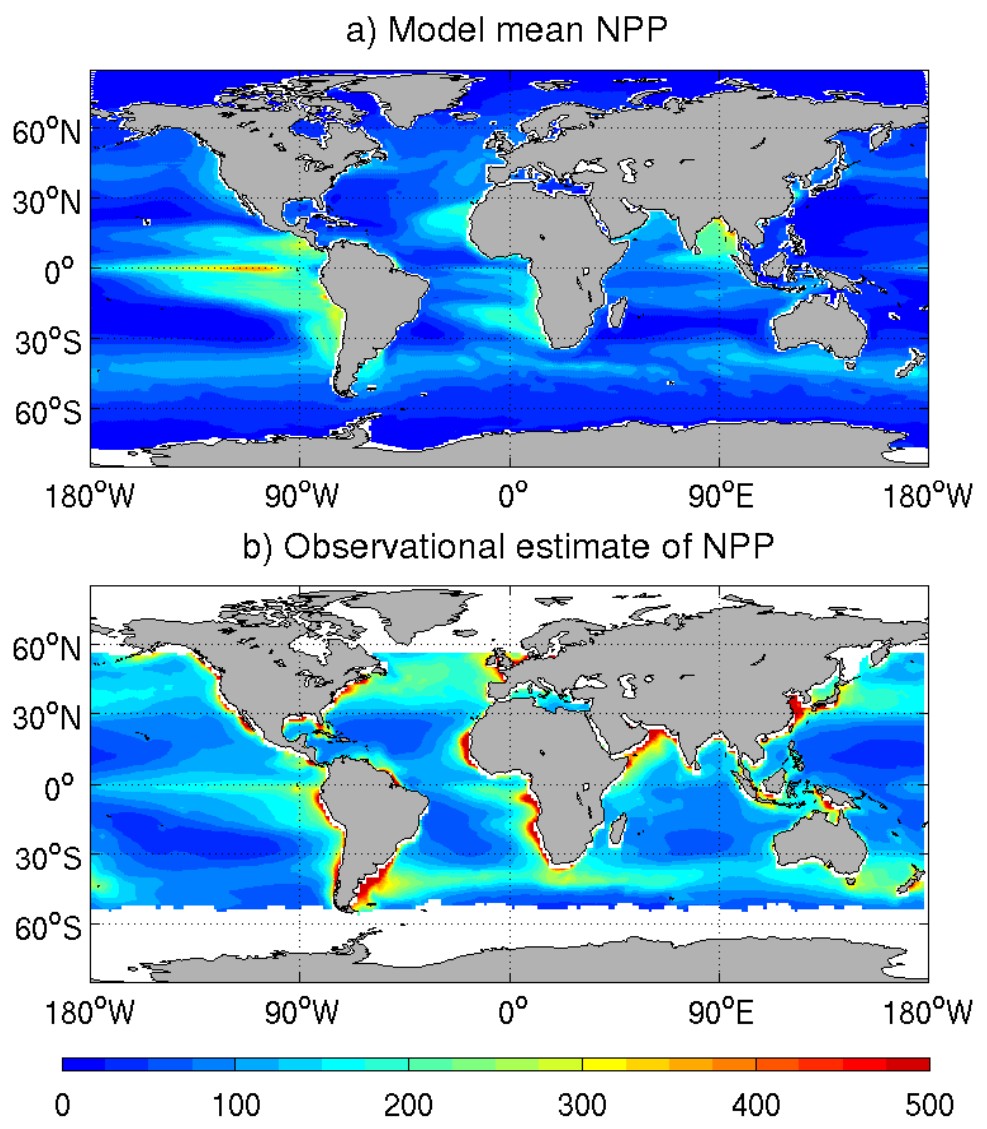

**Figure 9.** Spatial distribution of annual mean ocean primary production from a) model and b) observation. The observational estimate is based on remotely sensed chlorophyll data and the Vertically Generalized Production Model (VGPM) from Behrenfeld and Falkowski (1997). Model value is taken from the last 50 years of the PI control period. Units are in g C m$^{-2}$ yr$^{-1}$.





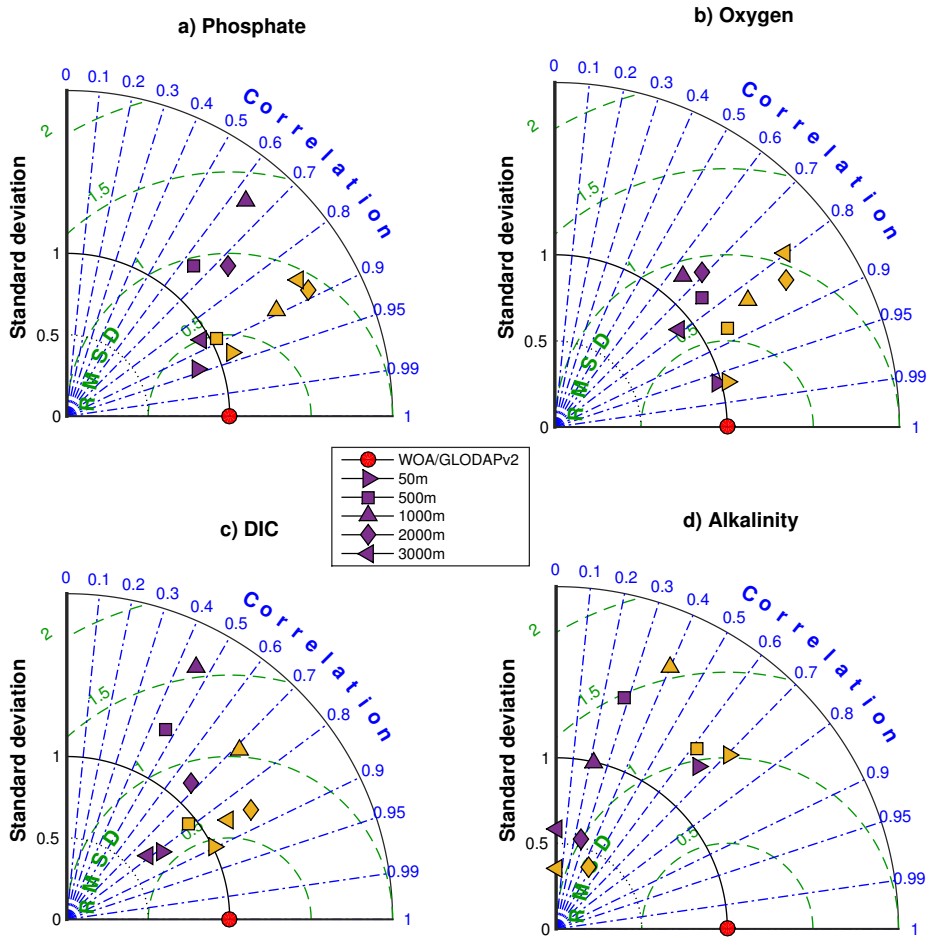

**Figure 10.** Taylor diagrams summarizing the statistical performance of the climatology fields of a) phosphate, b) oxygen, c) dissolved inorganic carbon, and d) total alkalinity in the water column relative to the observations (Garcia et al., 2014a, b; Lauvset et al., 2016). All standard deviations are normalized to the respective values from the observations. Different symbols represent model-data comparison at different depths from 50 m to 3000 m. Purple markers represent values from NorESM1-ME preindustrial simulations (Tjiputra et al., 2013), whereas yellow markers depict the NorESM1-F, and red circles indicate observations.





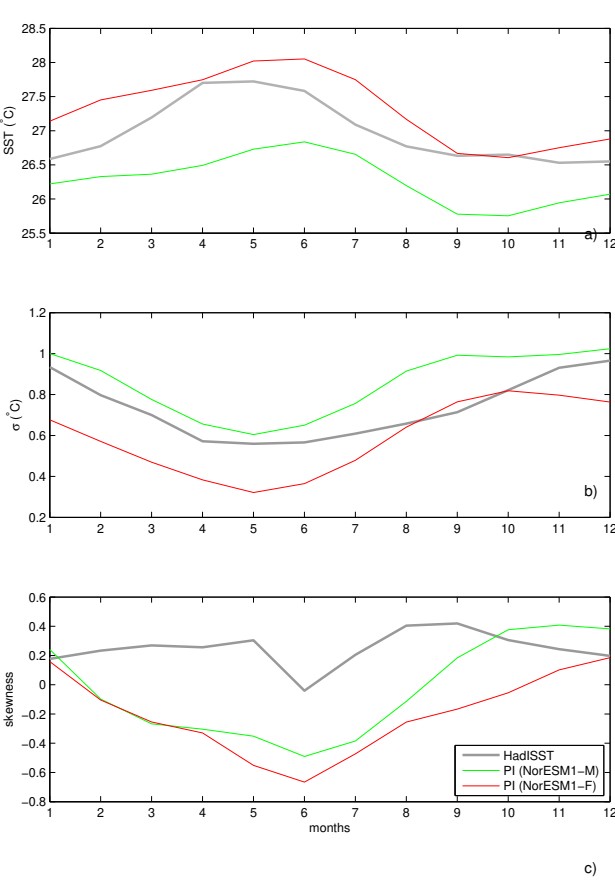

**Figure 11.** Monthly interannual a) SST, b) standard deviation of SST anomalies, and c) skewness of SST anomalies in the NINO3.4 region for the PI control experiments (last 500 years) of NorESM1-F and NorESM1-M. Data from HadISST (averaged between 1900 and 2005) are also plotted for comparison.



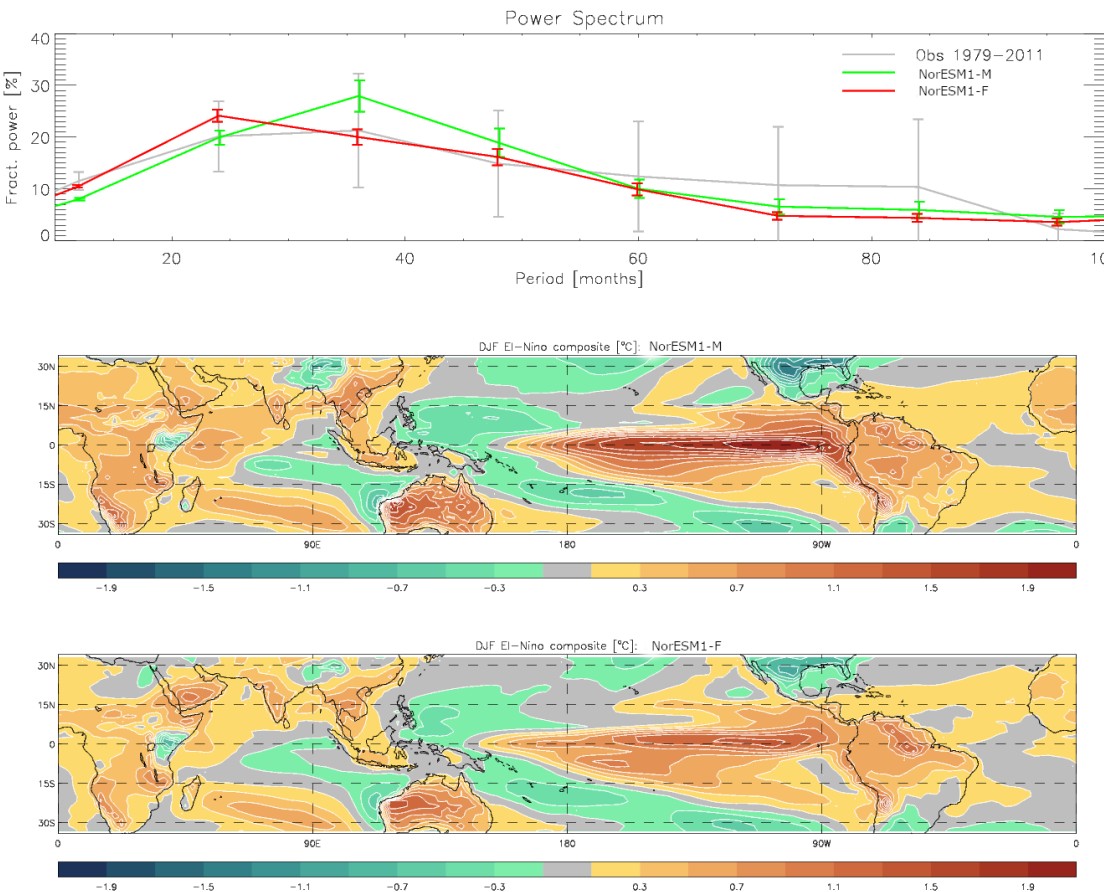

**Figure 12.** a) Power spectra of the NINO3.4 index for the PI control experiments of NorESM1-F and NorESM1-M, with comparison to HadISST data. Composite DJF surface temperature anomalies during El Niño years for b) NorESM1-M, and c) NorESM1-F. The data is from last 500 years of the PI control experiments for both models.





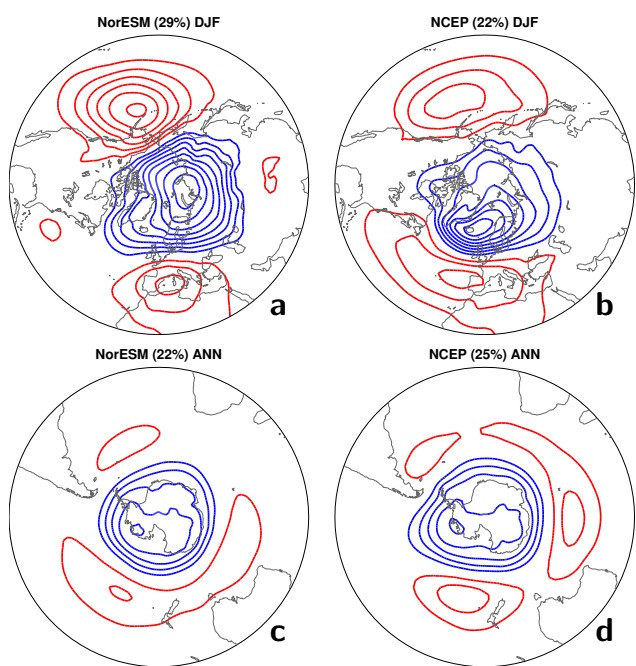

**Figure 13.** a,b) Leading empirical orthogonal function (EOF) of the winter (DJF) mean sea level pressure (SLP) anomalies over the NH (20-90° N) for a) the historical run (years 1976-2005) and b) NCEP-2 data of the same period. c,d) Leading EOF of the monthly mean SLP anomalies over the SH (90-20° S) for c) the historical run (years 1976-2005) and d) NCEP-2 data of the same period. The SLP patterns are obtained by regression of anomalies on the leading principal component time series. The contour intervals in all panels are 1 hPa, with the zero line omitted.





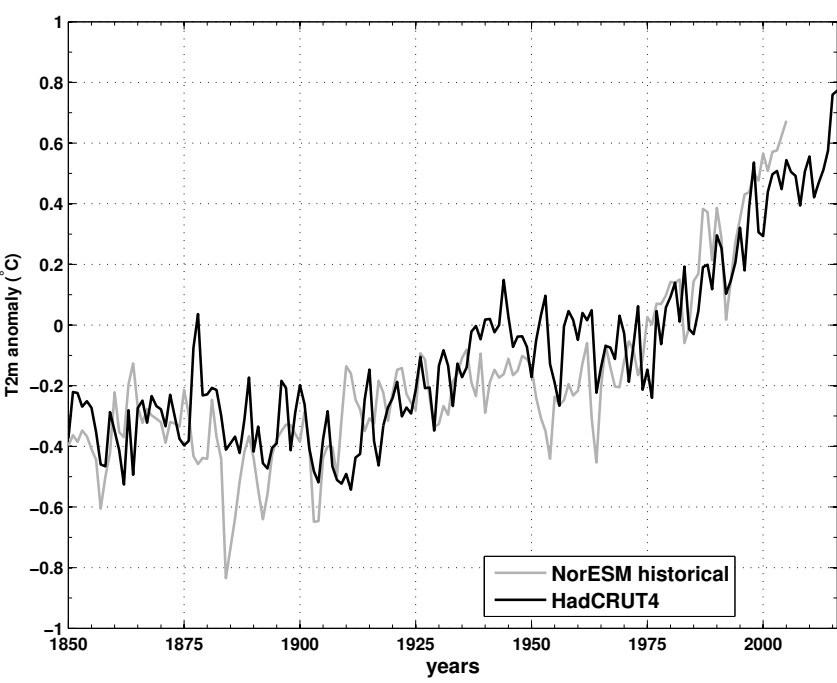

**Figure 14.** Modeled historical (grey line) and observed (black line) time series of global averaged surface temperature anomalies relative to the period of 1961-1990. Observations are derived from HadCRUT4 data set (Morice et al., 2012).





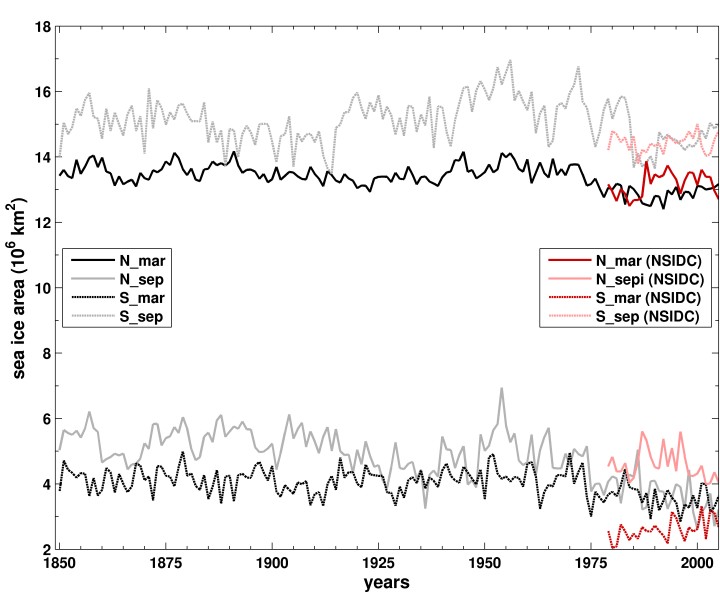

**Figure 15.** Modeled historical (black and grey lines) and observed (red lines) time series of sea ice area ($10^6$ km$^2$) in the Northern and Southern Hemispheres for September and March, respectively. Observations are from NSIDC (Fetterer et al., 2016).

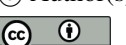



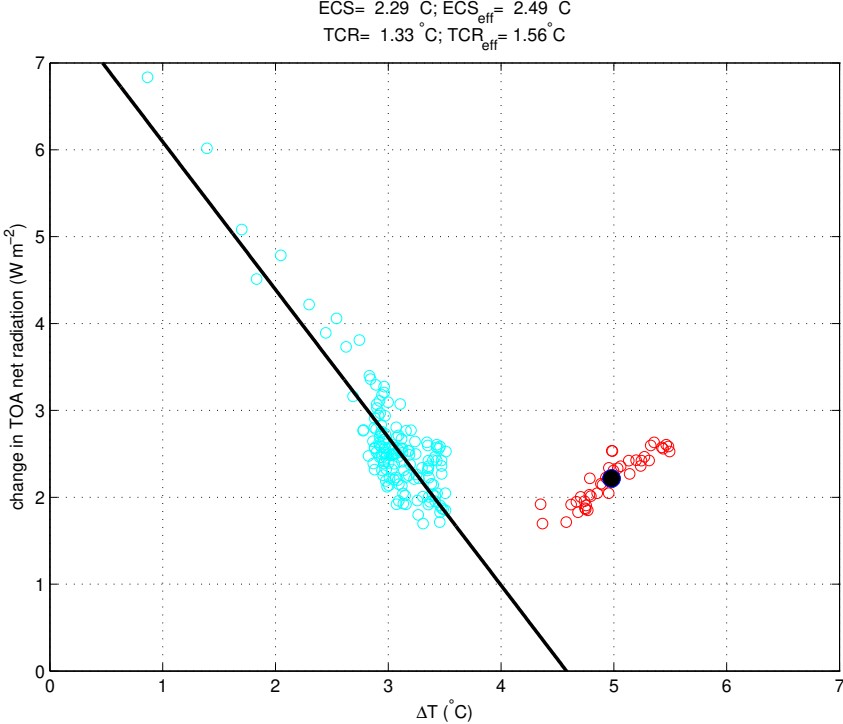

**Figure 16.** Estimation of effective climate sensitivity (ECS) and effective ECS ($ECS_{eff}$) in NorESM1-F. Blue dots are simulated change of TOA radiation flux $\Delta R$ (W m$^{-2}$) versus the change of global mean surface air temperature $\Delta T$ (°C) in the "abrupt $4\times CO_2$" experiment; the black line is the linear regression of the two variables. Red dots denote the effective temperature response in the last 40 years of the "abrupt $4\times CO_2$" experiment, and the black dot is the average (see text for further explanation).