# Peer review of "Description and evaluation of NorESM1-F: A fast version of the Norwegian Earth System Model (NorESM)"

_Geoscientific Model Development, 2018_

## Referee Comment (RC1) · D. Ivanova (Referee) · 22 Oct 2018

General comments The manuscript by Guo et al., presents a new version of the Norwegian Earth System Model (NorESM) featured as computationally fast and efficient with the main goal to be utilized in long-term paleoclimate simulations. The model efficiency is mainly achieved by using reduced complexity prescribed atmospheric chemistry as well as employing tripole grid in the ocean and sea ice domains which produces more stable solutions and allows longer time steps compared to the previous model version using dipole grid. The paper describes major model developments among which are energy formulation change in CAM4 physics, COARE-3 algorithm for air-sea flux

exchange, updated GM parameterization of eddy induced transport on neutral surfaces instead of isopycnal changing the vertical re-stratification, k-e turbulence closure scheme of second order parameterizing the diapycnal shear mixing substituting the previous K-profile vertical mixing scheme in the MICOM ocean component , more comprehensive particulate sinking scheme in the HAMMOC ocean carbon cycle component. The model fidelity is evaluated in terms of mean state, equilibrium, variability and climate sensitivity. Major model improvement compared to the previous model version is the more realistic AMOC.

Overall, this manuscript documents the major model developments and improvements in the NorESM models family which sets their entrance in the next Climate Model Intercomparison Program 6. I recommend publication after minor revisions.

Specific comments

- Weddell Sea polynya and Southern ocean deep convection

The authors have discussed improvement of the temperature and salinity biases in the intermediate layers and degrading of the SST and SSS biases in the surface layers, in the new model version NorESM-F compared to the previous NorESM-M. Since the goal is to use NorESM-F in a long term paleo simulations I think it is important to extend the model fidelity evaluation to the deep water formation and bottom water properties. Although, I do acknowledge the brief mentioning of AAWB production in the end of section 4.1 (lines 26-27, p.10)

Particularly, I found interesting the emergence of the Weddell Sea (WS) polynya in the new model version which doesn't seem to be evident in the earlier model version. This might be due to the improved sea ice simulation, but also due to the different vertical mixing and re-stratification representation. However, the occurrence of the Weddell Sea polynya signature in sea ice concentration/thickness September climatology (fig.8d) for the industrial period (1979-2005), when it was rarely observed (not present in the Hadley climatology), is a concern. It suggests that the WS polynya either

emerges too often or for too long periods. This in turn implies that the new model is producing deep water in the Southern Ocean (SO) with unrealistically intensified open water convection - common feature in the majority of the CMIP5 models.

On the other hand, earlier studies on the SO deep convection in CMIP5 models disagree about the convective behavior of the previous NorESM-M model. While de Lavergne et al (2014) and Heuzé et al (2013) classified NorESM-M as non-convective, Behrens et al (2016) using more comprehensive T-S analysis have shown that NorESM-M family may be classified as convective.

I recommend to add discussion in the current manuscript about the deep water convection in the NorESM-F and compare the new model to the previous by using some of the metrics in the published studies, e.g. showing the difference with WOCE climatology of the mean bottom potential density $\sigma 2$ and August mixed layer depth as in Heuzé et al (2013), see their Figure 2 or/and T-S diagram showing time mean ventilated volume as in Behrens et al (2016), see their Figure 6. The differences/similarities might also highlight the effect of some of the new model developments.

- Representation of the ice sheets/glaciers melt

Interactive ice sheet modeling is still under development in the current generation of fully-coupled climate models. Still the effect of the melting ice-sheets and glaciers can be important in a long term millennial simulations. Particularly, for more realistic representation of ocean re-stratification and water mass properties, as well as, for sea level rise implications. Is there any representation of the ice sheet/glaciers melt freshwater/heat/volume flux in NorESM-F model? If yes, can you please add discussion in the manuscript.

Technical corrections

- p. 2, l. 10 For completeness and quick comparison could you state the resolution and performance metrics of NorESM-L. - p.4, l.11-15 Can you also state ocean time step

and coupled frequency - p2, l. 26 Typo "fugure" instead of "future" - p.4, l.31-32 Please reword the last two sentences on p.4 Maybe as: The implementation of this algorithm improved the evaporation-wind stress relationship, which appears too steep in CAM4 compared to observations (see supplementary Fig.S3) - Fig.8 Sea ice plots – enlarge – panel of 2x2 - I wasn't able to see the supplemental material. I dowloaded the .zip archive from the website but when I unzipped it on my Mac, the system didn't recognize it. All I know it is a single binary file which is neither executable or readable.

References

Behrens, E., G. Rickard, O. Morgenstern, T. Martin, A. Osprey, and M. Joshi (2016), Southern Ocean deep convection in global climate models: A driver for variability of subpolar gyres and Drake Passage transport on decadal timescales, J. Geophys. Res. Oceans, 121, 3905–3925, doi:10.1002/ 2015JC011286. Heuzé, C., K. J. Heywood, D. P. Stevens, and J. K. Ridley (2013), Southern Ocean bottom water characteristics in CMIP5 models, Geophys. Res. Lett., 40, 1409–1414, doi:10.1002/grl.50287
* * *

---

## Referee Comment (RC2) · Anonymous Referee #2 · 27 Nov 2018

General comments The manuscript by Guo et al. presents a new version of the Norwegian Earth System Model, i.e., NorESM1-F, that is designed for millennium-scale and large ensemble climate simulations. The paper describes the major developments of the model from its predecessor, NorESM1-M. These developments lead to substantial improvement of the computational efficiency, and better representations of the atmospheric and oceanic physics as well as ocean biogeochemistry. The model performance is documented by examination of the equilibrium state of a 2000 year spin-up and control run forced with pre-industrial conditions, and evaluation of the model transient climate using observations and NorESM1-M as benchmarks. In general the model shows a satisfactory equilibrium with only a slightly cooling trend in 1000 years,

and a good agreement with the observational estimates of the present day climate state. In comparison with the NorESM1-M, the new model demonstrates comparable or reduced biases. A particular feature of the improvements lays in the much more realistic strength of the simulated meridional overturning circulation, which results in more realistic atmospheric heat transport in the Atlantic Ocean and reduction of the warm and saline bias in the deep Atlantic. The more realistic physical ocean consequently improves the simulated interior ocean biogeochemical tracers.

Overall the manuscript is well written, and clearly documents the major development and performance of the new model version of the NorESM. As large ensemble has become an important way forward in understanding climate variability and quantifying climate change projections, I believe the NorESM1-F with its computational efficiency, will make important contributions to studies of the millennium-scale climate change as well as to the Coupled Model Intercomparison project 6. I recommend publication in GMD subject to the following minor (and mostly technical) revisions.

Specific comments • The authors demonstrate that the simulated Atlantic meridional overturning circulation (AMOC) in NorESM1-F is improved greatly and is much more realistic in comparison with NorESM1-M. This is a very encouraging improvement. As getting a realistic AMOC is often a difficult task in climate modeling, and to my knowledge, it is also a long standing problem in NorESM models. It is thus worth to discuss which model developments lead to such an achievement. This is potentially important for future model development. Minor comments • Page 2, line 32-32: What is the vertical resolution of the atmosphere and ocean component of the NorESM1-F? It is not mentioned in the manuscript. These can be state here, where the horizontal resolutions are given. • Page 2, line 34-35: do you really ran the model configured with the biogeochemistry using less cores than the model with the biogeochemistry deactivated? This doesn't sound logic to me. • Page 4, line 23: here "thus" should be "that"? • Page 10, line 23: "cleanly" should be "clearly". • Page 10, line 28: what is AABW stands for? • Page 14, line 28: "the" should be deleted. • Page

27, figure 6: I suggest to add zero lines in the figures to increase the readability of the figures.
* * *

---

## Author Comment (AC1) · 14 Dec 2018

Response to Detelina Ivanova:

We respond to the referee's comments in blue font below.

General comments
The manuscript by Guo et al., presents a new version of the Norwegian Earth System Model (NorESM) featured as computationally fast and efficient with the main goal to be utilized in long-term paleoclimate simulations. The model efficiency is mainly achieved by using reduced complexity prescribed atmospheric chemistry as well as employing tripole grid in the ocean and sea ice domains which produces more stable solutions and allows longer time steps compared to the previous model version using dipole grid. The paper describes major model developments among which are energy formulation change in CAM4 physics, COARE-3 algorithm for air-sea flux exchange, updated GM parameterization of eddy induced transport on neutral surfaces instead of isopycnal changing the vertical re-stratification, k-e turbulence closure scheme of second order parameterizing the diapycnal shear mixing substituting the previous K-profile vertical mixing scheme in the MICOM ocean component , more comprehensive particulate sinking scheme in the HAMMOC ocean carbon cycle component. The model fidelity is evaluated in terms of mean state, equilibrium, variability and climate sensitivity. Major model improvement compared to the previous model version is the more realistic AMOC. Overall, this manuscript documents the major model developments and improvements in the NorESM models family which sets their entrance in the next Climate Model Intercomparison Program 6. I recommend publication after minor revisions.

We thank Detelina Ivanova for the assessment and overall positive comments on our manuscript. We respond to the specific comments below point by point.

Specific comments
- Weddell Sea polynya and Southern ocean deep convection

The authors have discussed improvement of the temperature and salinity biases in the intermediate layers and degrading of the SST and SSS biases in the surface layers, in the new model version NorESM-F compared to the previous NorESM-M. Since the goal is to use NorESM-F in a long term paleo simulations I think it is important to extend the model fidelity evaluation to the deep water formation and bottom water properties. Although, I do acknowledge the brief mentioning of AAWB production in the end of section 4.1 (lines 26-27, p.10)

Particularly, I found interesting the emergence of the Weddell Sea (WS) polynya in the new model version which doesn't seem to be evident in the earlier model version. This might be due to the improved sea ice simulation, but also due to the different vertical mixing and re-stratification representation. However, the occurrence of the Weddell Sea polynya signature in sea ice concentration/thickness September climatology (fig.8d) for the industrial period (1979-2005), when it was rarely observed (not present in the Hadley climatology), is a concern. It suggests that the WS polynya either emerges too often or for too long periods. This in turn implies that the new model is producing deep water in the Southern Ocean (SO) with unrealistically intensified open water convection - common feature in the majority of the CMIP5 models.

On the other hand, earlier studies on the SO deep convection in CMIP5 models disagree about the convective behavior of the previous NorESM-M model. While de Lavergne et al (2014) and Heuzé et al (2013) classified NorESM-M as nonconvective, Behrens et al (2016) using more comprehensive T-S analysis have shown that NorESM-M family may be classified as convective. I recommend to add discussion in the current manuscript about the deep water convection in the NorESM-F and compare the new model to the previous by using some of the metrics in the published studies, e.g. showing the difference with WOCE climatology of the mean bottom potential density $\sigma_2$ and August mixed layer depth as in Heuzé et al (2013), see their Figure 2 or/and T-S

diagram showing time mean ventilated volume as in Behrens et al (2016), see their Figure 6. The differences/similarities might also highlight the effect of some of the new model developments.

We share the same interest and concern with the reviewer on the Weddell Sea polynya and Southern Ocean deep convection.

In both the PI and historical NorESM1-F simulations, Weddell Sea Polynyas seem to be a persistent feature, although their sizes are generally much smaller than the one shown in Fig. S5, and their area and location vary from year to year. The sizes of the polynyas show a tendency to become larger in the last few decades of the twentieth century, which is inconsistent with de Lavergne et al. (2014) who showed that simulated polynyas in CMIP5 models tend to cease during the global warming period due to the fresher surface in the Southern Ocean. However, the September mixed layer depths (MLD; Fig. 1) in the Southern Ocean are overall shallower in NorESM1-F compared to that in NorESM1-M, except in the Weddell Sea polynya region and another region in the Indian sector of the Southern Ocean. Further analysis of Southern Ocean zonal mean T/S/ideal age profiles (Figs. 2 & 3) shows a colder (0.5-1 degC) and less ventilated deep water in NorESM1-F relative to NorESM1-M, indicating an overall more stratified and less convected Southern Ocean in NorESM1-F, consistent with the MLD analysis.

Given the already considerable length of the manuscript, and an intention to make a balance among the different sections, we would prefer to include the lower panel of Fig. 1 and Fig. 3 in supplementary material. Following the reviewer's comments, we add more discussion and rephrase the original P10, L26-27 accordingly as follows:

"The production of AABW in the Atlantic sector is enhanced during the deep convection process, with an increase in volume transport of ~2.5 Sv at 30∘S in the South Atlantic. Apart from the two dramatic events

described above, Weddell Sea polynyas are a persistent feature (albeit with much smaller size) in both the PI and historical simulations of NorESM1-F, in contrast to NorESM1-M that does not show any sign of polynyas. However, analysis of the September mixed layer depth and ideal age (see supplementary Fig. S6 & S7) indicates that open ocean convection is overall reduced in the Southern Ocean in NorESM1-F compared to NorESM1-M, except in the Weddell Sea polynya region and a region in the Indian section of the Southern Ocean. A thinner sea ice in NorESM1-F relative to NorESM1-M is expected to be favorable for the occurrence of Weddell Sea polynyas. Additionally, the different vertical mixing schemes and mixed layer restratification in NorESM1-F compared to NorESM1-M are also likely to play a role in creating the polynyas, and also in the overall reduction of convection in the Southern Ocean."

[Figure]

Fig 1. Global map of September mixed layer depth (defined as $\sigma_0(z) - \sigma_0(10\,\text{m}) = 0.03$ kg m$^{-3}$ according to de Boyer Montégut et al., 2004) for NorESM1-F (upper panel), NorESM1-M (middle panel), and NorESM1-F minus NorESM1-M (lower panel). The calculations are based on the 50 year mean PI experiments for both model versions.

[Figure]

Fig 2. Southern Ocean zonal mean anomalies (NorESM1-F minus NorESM1-M) of potential temperature (left panel) and salinity (right panel). The calculations are based on the 50 year mean PI experiments for both model versions.

[Figure]

Fig 3. Southern Ocean zonal mean ideal age for NorESM1-M and NorESM1-F.

- Representation of the ice sheets/glaciers melt

Interactive ice sheet modeling is still under development in the current generation of fully-coupled climate models. Still the effect of the melting ice-sheets and glaciers can be important in a long term millennial simulations. Particularly, for more realistic representation of ocean re-stratification and water mass properties, as well as, for sea level rise implications. Is there any representation of the ice sheet/glaciers melt freshwater/heat/volume flux in NorESM-F model? If yes, can you please add discussion in the manuscript.

NorESM1-F does not have an interactive land ice component that accounts for changing rates of glacial melting, although we do acknowledge that such a capability would be of high interest and relevance for long past and future simulations. We are therefore currently exploring the feasibility of enabling the ice sheet component in NorESM.

Technical corrections
- p. 2, l. 10 For completeness and quick comparison could you state the resolution and performance metrics of NorESM-L.

We have stated in the manuscript that "NorESM-L employs a similar grid resolution as the lower-resolution CCSM4...", and the lower-resolution CCSM4 is mentioned in the previous paragraph. We therefore update the text to "NorESM-L employs a similar grid resolution as the lower-resolution CCSM4 mentioned above, with a throughput of ~50 model years per day with ~150 cores. NorESM-L has been used for ...".

- p.4, l.11-15 Can you also state ocean time step and coupled frequency

We add the following to the manuscript: "Compared to the bipolar grid, the tripolar grid is more isotropic at high northern latitudes and for comparable resolution allows an almost doubled time integration step for

the ocean component, e.g. from 1800 to 3200 seconds for the baroclinic time step."

The ocean component is coupled once a day with the rest of the components. But since this section is on "measures to improve computational efficiency", we would not include information on the ocean coupled frequency here.

- p2, l. 26 Typo "fugure" instead of "future"

corrected.

- p.4, l.31-32 Please reword the last two sentences on p.4 Maybe as: The implementation of this algorithm improved the evaporation-wind stress relationship, which appears too steep in CAM4 compared to observations (see supplementary Fig.S3)

Suggestion adopted.

- Fig.8 Sea ice plots – enlarge – panel of 2x2

Suggestion adopted.

- I wasn't able to see the supplemental material. I dowloaded the .zip archive from the website but when I unzipped it on my Mac, the system didn't recognize it. All I know it is a single binary file which is neither executable or readable.

The journal staff contacted us regarding this technical error with the supplementary material. They have helped solve the problem and the supplementary material can be downloaded and opened properly now.

---

## Author Comment (AC2) · 14 Dec 2018

Response to Anonymous Referee #2:

We respond to the referee's comments in blue font below.

General comments
The manuscript by Guo et al. presents a new version of the Norwegian Earth System Model, i.e., NorESM1-F, that is designed for millennium-scale and large ensemble climate simulations. The paper describes the major developments of the model from its predecessor, NorESM1-M. These developments lead to substantial improvement of the computational efficiency, and better representations of the atmospheric and oceanic physics as well as ocean biogeochemistry. The model performance is documented by examination of the equilibrium state of a 2000 year spinup and control run forced with pre-industrial conditions, and evaluation of the model transient climate using observations and NorESM1-M as benchmarks. In general the model shows a satisfactory equilibrium with only a slightly cooling trend in 1000 years, and a good agreement with the observational estimates of the present day climate state. In comparison with the NorESM1-M, the new model demonstrates comparable or reduced biases. A particular feature of the improvements lays in the much more realistic strength of the simulated meridional overturning circulation, which results in more realistic atmospheric heat transport in the Atlantic Ocean and reduction of the warm and saline bias in the deep Atlantic. The more realistic physical ocean consequently improves the simulated interior ocean biogeochemical tracers. Overall the manuscript is well written, and clearly documents the major development and performance of the new model version of the NorESM. As large ensemble has become an important way forward in understanding climate variability and quantifying climate change projections, I believe the NorESM1-F with its computational efficiency, will make important contributions to studies of the millennium-scale climate change as well as to the Coupled Model Intercomparison project 6. I recommend publication in GMD subject to the following minor (and mostly technical) revisions.

We thank the reviewer for his/her assessment and overall positive comments on our manuscript. We respond to the specific and minor comments below point by point.

Specific comments
The authors demonstrate that the simulated Atlantic meridional overturning circulation (AMOC) in NorESM1-F is improved greatly and is much more realistic in comparison with NorESM1-M. This is a very encouraging improvement. As getting a realistic AMOC is often a difficult task in climate modeling, and to my knowledge, it is also a long standing problem in NorESM models. It is thus worth to discuss which model developments lead to such an achievement. This is potentially important for future model development.

We agree with the reviewer that reducing the strength of AMOC in NorESM1-F is an encouraging improvement. We have stated in the manuscript that (P8, L28-30) "Contributing to the reduced AMOC in NorESM1-F is reduced deep convection in the Labrador Sea due to stronger upper ocean restratification by the reformulated GM and modified parameterization of ocean mixed layer restratification by submesoscale eddies."

Minor comments
Page 2, line 32-32: What is the vertical resolution of the atmosphere and ocean component of the NorESM1-F? It is not mentioned in the manuscript. These can be state here, where the horizontal resolutions are given.

We thank the reviewer for pointing this out. We add the following sentence - "There are 26 vertical levels in the atmosphere and 53 vertical layers in the ocean component, respectively."

Page 2, line 34-35: do you really ran the model configured with the biogeochemistry using less cores than the model with the biogeochemistry deactivated? This doesn't sound logic to me.

We actually carried out the production run on an older HPC that is scheduled to be depreciated soon. The reported model throughput in the manuscript is the *test speed* on the new HPC "FRAM" that we will be using.

Page 4, line 23: here "thus" should be "that"?

We think that "thus" is OK here.

Page 10, line 23: "cleanly" should be "clearly".

adopted.

Page 10, line 28:  what is AABW stands for?

AABW stands for Antarctic Bottom Water. We have replaced AABW with its full name in the revised manuscript.

Page 14, line 28: "the" should be deleted.

Yes, indeed.

Page27, figure 6: I suggest to add zero lines in the figures to increase the readability of the figures.

We have added the zero lines in the updated figure.

---

## Editor Comment (EC1) · Wang (Editor) · 17 Dec 2018

Dear Dr. Guo,

The fontsize in many figures, in particular in Fig. 2 and 4, is too small. Please check all your figures including those in the SI to make sure the text in figures has proper fontsize. In addition, the text in Fig. 12 appears to be gray in the current version.

Best wishes

Qiang Wang

---

## Author Comment (AC3) · 21 Dec 2018

We thank the Editor for the comments. Following the Editor's request, we have increased the font size in Figs. 2, 3, 4, 11, 14, 15, 16, and in supplementary Figs. 5, 7.

---

## Author Response (AR2)

**Response to the Editor's comments**

We thank the Editor for the comments. Following the Editor's request, we have increased the font size in Figs. 2, 3, 4, 11, 14, 15, 16, and in supplementary Figs. 5, 7.